# Meta-Complementing the Semantics of Short Texts in Neural Topic Models

**Delvin Ce Zhang**
School of Computing and Information Systems
Singapore Management University
Singapore 178902
cezhang.2018@smu.edu.sg

**Hady W. Lauw**
School of Computing and Information Systems
Singapore Management University
Singapore 178902
hadywlauw@smu.edu.sg

## Abstract

Topic models infer latent topic distributions based on observed word co-occurrences in a text corpus. While typically a corpus contains documents of variable lengths, most previous topic models treat documents of different lengths uniformly, assuming that each document is sufficiently informative. However, shorter documents may have only a few word co-occurrences, resulting in inferior topic quality. Some other previous works assume that all documents are short, and leverage external auxiliary data, e.g., pretrained word embeddings and document connectivity. Orthogonal to existing works, we remedy this problem within the corpus itself by proposing a Meta-Complement Topic Model, which improves topic quality of short texts by transferring the semantic knowledge learned on long documents to complement semantically limited short texts. As a self-contained module, our framework is agnostic to auxiliary data and can be further improved by flexibly integrating them into our framework. Specifically, when incorporating document connectivity, we further extend our framework to complement documents with limited edges. Experiments demonstrate the advantage of our framework.

## 1 Introduction

Much of the data on the Web can be represented as text documents. Topic models help to understand the main themes within documents, i.e., each document is represented by a topic distribution, and each topic is interpreted by its key words. The quality of topic distribution of each document depends on sufficient word co-occurrences. However, many real-word corpora contain documents of *variable lengths*. Academic papers vary from journal manuscripts to conference papers to extended abstracts. News articles could be headlines, short or full articles, or detailed commentaries. Fig. 1(a) illustrates an academic paper corpus where the distribution of document lengths exhibits a long-tail distribution. Despite variable lengths (with different degrees of sufficiency of word co-occurrences), existing works, e.g., ProdLDA [31] and GATON [37], treat documents uniformly, resulting in inferior topic quality for short texts. Evidentially, Fig. 1(b) presents document classification accuracy on four subsets of corpus with descending lengths. Accuracies gradually drop as the length decreases. The inferior topic quality of short texts limits the overall performance of a topic model.

36th Conference on Neural Information Processing Systems (NeurIPS 2022).

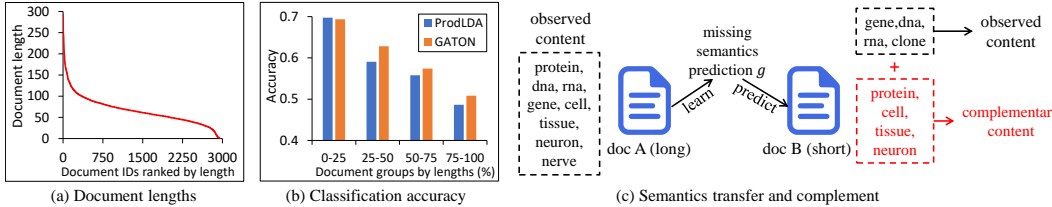

Figure 1: Illustration of (a) a paper corpus with various-length documents, (b) classification accuracy on four subsets of the corpus by descending length, and (c) semantic transfer and complement.

**Challenges.** Most existing topic models optimize the learning process by averaging the generative losses of different documents, without paying special attention to semantically limited short texts. A few studies, e.g., OTLDA [15], take weighted summation of losses based on document lengths, they compute the weights by dividing the length of each document by the length of the whole corpus, which further deemphasizes the importance of short text modeling. Thus, we seek to improve short text topic modeling within a *variable-length* corpus, without hurting topic quality of long documents.

To mitigate the scarcity of word co-occurrences in short texts, some works leverage auxiliary knowledge to enhance topic modeling. ETM [8] exploits pretrained word embeddings [27, 29] to capture word similarities. RTM [5] constructs a document network, e.g., paper citation network, to aggregate topics of connected documents. However, they rely on the availability of auxiliary data.

**Approach.** We propose **M**eta-**C**omplement **T**opic **M**odel (MCTM). Since a corpus contains variable lengths of documents, we are motivated to learn the transferable semantic knowledge on long documents, and complement the semantics-scarce short texts and enhance the latter's topic distributions. As illustrated by Fig. 1(c), we learn a function $g$ on long document $A$ with abundant content, and use it to predict missing semantics (red box) for short document $B$ to complement its content for a more expressive topic distribution. Since meta-learning [10] is emerging to improve model performance with few labeled observations, but no one explores its design in topic modeling. We are thus motivated to integrate it and optimize the proposed MCTM by a meta-learning objective.

Orthogonal to existing works relying on auxiliary data, our framework is self-contained, assuming only in-corpus information, which offers a new direction to improve short text topic modeling. When auxiliary data are available, our framework can be further improved by flexibly incorporating them. In particular, when incorporating document network structure, we discover that document degrees also exhibit a similar long-tail distribution, i.e., some structure-abundant documents link to sufficient neighbors as auxiliary, while others contain scarce links. Besides *textual* semantic complement at Fig. 1(c), we extend MCTM to further complement *structural* semantics on the document network.

**Contributions.** Our contributions are as follows. *First*, we propose MCTM, which learns how to complement textual semantics by semantic knowledge transfer. *Second*, we derive two alternatives to implement missing semantics prediction function $g$ to capture document similarities. *Third*, although agnostic to auxiliary data, MCTM can also flexibly integrate them to further improve the performance. We demonstrate our adaptability by modeling pretrained word embeddings and document networks. For the latter, we extend MCTM to further complement structural semantics. *Fourth*, extensive experiments verify the effectiveness of MCTM.

## 2 Problem Formulation and Preliminaries

We are given a corpus of $N$ documents $\mathcal{D} = \{\mathbf{d}_i\}_{i=1}^N$. Each document $\mathbf{d} \in \mathbb{R}^{|\mathcal{V}|}$ is a vector in the vocabulary space $\mathcal{V}$. $l_d = \sum_{w \in \mathcal{V}} d_w$ is the length of document $d$ where $d_w$ is the word count of $w$ in $d$. When word embeddings are available, we have $\mathcal{H} = \{\mathbf{h}_w\}_{w \in \mathcal{V}}$ where $\mathbf{h}_w$ is the embedding of word $w$. Documents may link to others in a document network $\mathcal{G} = \{\mathcal{D}, \mathcal{E}\}$, with documents $\mathcal{D}$ and network connectivity $\mathcal{E} = \{e_{ij}\}_{i,j=1}^N$. If document $d$ links to $d'$, $e_{d,d'} = 1$, otherwise $e_{d,d'} = 0$. $\mathcal{N}(d)$ is the set of neighbors of $d$. We consider an undirected network, $e_{d,d'} = e_{d',d}$. Corpus $\mathcal{D}$ contains documents of variable lengths $\{l_i\}_{i=1}^N$. We introduce a hyperparameter $\mathscr{L}$ as the threshold where short documents are those with fewer observed words, $\mathcal{D}_{\text{short}} = \{\mathbf{d}_i | l_i < \mathscr{L}\}$, and long documents are defined symmetrically, $\mathcal{D}_{\text{long}} = \{\mathbf{d}_i | l_i \geq \mathscr{L}\}$. We consider $\mathscr{L}$ as a predefined hyperparameter and leave other designs as future work. See Table 1 for a summary of math notations.

Table 1: Summary of math notations.

| Notation | Description |
|---|---|
| $\mathcal{D}$ | a corpus of $N$ documents, $\mathcal{D} = \{\mathbf{d}_i\}_{i=1}^N$ |
| $\mathbf{d}$ | a document representation in the word space, $\mathbf{d} \in \mathbb{R}^{|\mathcal{V}|}$ |
| $\mathcal{V}$ | vocabulary |
| $l_d$ | length of document $d$ |
| $\mathcal{H}$ | a set of pretrained word embeddings |
| $\mathbf{h}_w$ | pretrained word embedding of word $w$ |
| $\mathcal{G}$ | a document network |
| $\mathcal{E}$ | a set of links connecting documents |
| $\mathcal{N}(d)$ | the neighbor set of document $d$ |
| $\mathcal{D}_{\text{long}}$ | the subset of long documents in corpus $\mathcal{D}$ |
| $\mathcal{D}_{\text{short}}$ | the subset of short documents in corpus $\mathcal{D}$ |
| $\mathcal{T}_d$ | the task of document $d$, i.e., generating observed words of document $d$ |
| $\mathcal{S}_d$ | a set of support words of document $d$ |
| $\mathcal{Q}_d$ | a set of query words of document $d$ |
| $\theta$ | a collection of parameters of encoder $f_\theta$ |
| $\mathbf{z}_d$ | topic distribution of document $d$, $\mathbf{z}_d \in \mathbb{R}^K$ |
| $K$ | number of topics |
| $\mathbf{m}_d$ | missing semantics of document $d$ |
| $\mu$ | a collection of parameters of semantics prediction function $g_\mu$ |
| $\mathcal{S}_{\mathcal{N}(d)}$ | a set of support neighbors of document $d$ |
| $\mathcal{Q}_{\mathcal{N}(d)}$ | a set of query neighbors of document $d$ |

Given a variable-length corpus $\mathcal{D}$ (as well as word embeddings $\mathcal{H}$ and network $\mathcal{E}$ if observed) as input, we aim to output topic distributions for documents where the topic quality of short documents is improved, without hurting long documents. Note that our goal is not to allow short texts to reach the performance of long documents, but to improve short text topic modeling as much as possible.

**Meta-Learning.** Meta-learning [10] optimizes globally shared parameters, a.k.a. prior knowledge, over meta-training tasks, so as to rapidly adapt the model to previously unseen meta-testing tasks with only a few observed data. Since topic models generally learn topics by a content generative process, here we consider generating observed words for a document $d$ as a task $\mathcal{T}_d$. A meta-training task $\mathcal{T}_d$ corresponds to a training document $d$ and consists of a support set and a query set, $\mathcal{T}_d = \{\mathcal{S}_d, \mathcal{Q}_d\}$. Each set contains randomly sampled words from document $d$, such that support words and query words are mutually exclusive, $\mathcal{S}_d \cap \mathcal{Q}_d = \emptyset$ and $\mathcal{S}_d \cup \mathcal{Q}_d = \mathbf{d}$. *Meta-training* has two steps:

1. Local update. Given a topic model $f_\theta$ with parameter $\theta$, $f_\theta$ is first updated from the globally shared parameter $\theta$ to document-specific local parameter $\theta_d$ w.r.t. loss on $d$'s support words $\mathcal{S}_d$.

$$\theta_d = \theta - \alpha \nabla_\theta \mathcal{L}(\theta, \mathcal{S}_d) \quad \text{where} \quad \mathcal{T}_d = \{\mathcal{S}_d, \mathcal{Q}_d\} \in \mathcal{T}_{\text{tr}}. \tag{1}$$

$\alpha$ is meta-learning rate, $\nabla$ is gradient, $\mathcal{L}$ is loss function, $\mathcal{T}_{\text{tr}}$ is a set of meta-training tasks (documents).

2. Global update. After obtaining $\theta_d$ for each document $d$, we compute the loss on query words $\mathcal{L}(\theta_d, \mathcal{Q}_d)$. Together with other training tasks, we optimize the globally shared parameter $\theta$.

$$\theta^* = \min_\theta \sum_{\mathcal{T}_d \in \mathcal{T}_{\text{tr}}} \mathcal{L}(\theta_d, \mathcal{Q}_d) = \min_\theta \sum_{\mathcal{T}_d \in \mathcal{T}_{\text{tr}}} \mathcal{L}(\theta - \alpha \nabla_\theta \mathcal{L}(\theta, \mathcal{S}_d), \mathcal{Q}_d). \tag{2}$$

$\theta^*$ is the new globally shared parameter and will replace $\theta$ at Eq. 1–2 for the next iteration. After convergence, the final global parameter $\theta^*$ can easily be adapted to meta-testing tasks.

*Meta-testing* tasks $\mathcal{T}_{\text{te}}$ are unseen test documents. All the observed words are support words, $\mathcal{T}_d = \mathcal{S}_d = \mathbf{d}$. During meta-testing, topic model $f_{\theta^*}$ with optimized global parameter $\theta^*$ is updated w.r.t. $\mathcal{S}_d$ by Eq. 1 and obtain $\theta_d^*$. The topic distribution of testing document is inferred by $\mathbf{z}_d = f_{\theta_d^*}(\mathbf{d})$.

## 3  Methodology

We introduce **M**eta-**C**omplement **T**opic-**M**odel (MCTM) at Fig. 2. Below we elaborate three components, graph convolutional encoder, missing semantics prediction, and meta-learning optimization.

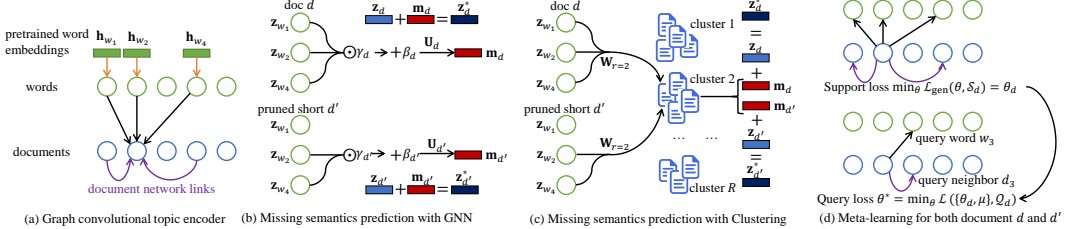



(a) Graph convolutional topic encoder    (b) Missing semantics prediction with GNN    (c) Missing semantics prediction with Clustering    (d) Meta-learning for both document $d$ and $d'$

Figure 2: Model architecture of Meta-Complementing Topic Model, MCTM.



## 3.1 Graph Convolutional Topic Encoding

We follow GATON [37] and first present a graph convolutional encoder $f_\theta$ (Fig. 2(a)), which projects documents to $K$-dimensional topic distributions. We defer the discussion on short text modeling to the following subsections. Given a corpus $\mathcal{D}$, considering documents and words as vertices, we construct a bipartite graph, the links represent word occurrences in the documents. Since documents and words preserve heterogeneous feature spaces, we project them to the same topic space by learnable matrix,

$$\tilde{\mathbf{z}}_d^{(l+1)} = \mathbf{W}_1^{(l+1)}\mathbf{z}_d^{(l)}, \qquad \tilde{\mathbf{z}}_w^{(l+1)} = \mathbf{W}_2^{(l+1)}\mathbf{z}_w^{(l)}, \quad \text{where} \quad d \in \mathcal{D}, w \in \mathcal{V}. \tag{3}$$

$l$ is the $l$-th convolutional layer. $\mathbf{z}_d^{(0)}$ and $\mathbf{z}_w^{(0)}$ are inputs, i.e., Bag-of-Words and one-hot, respectively.

To differentiate the importance of words, we evaluate attention between document $d$ and its observed support words at Eq. 4 and aggregate words at Eq. 5 where $[\cdot||\cdot]$ is concatenation.

$$a_{d,w} = \text{softmax}\Big(\tanh(\mathbf{b}_1^{(l+1)\top}[\tilde{\mathbf{z}}_d^{(l+1)}||\tilde{\mathbf{z}}_w^{(l+1)}])\Big) \quad \text{where} \quad w \in \mathcal{S}_d \tag{4}$$

$$\mathbf{z}_d^{(l+1)} = \tanh\Big(\frac{1}{2}(\tilde{\mathbf{z}}_d^{(l+1)} + \sum_{w \in \mathcal{S}_d} a_{d,w}\tilde{\mathbf{z}}_w^{(l+1)})\Big). \tag{5}$$

Symmetrically, we modify Eq. 4 and obtain $a_{w,d}$, i.e., the attention between word $w$ and documents it appears in. After symmetric aggregation at Eq. 5, we obtain $\mathbf{z}_w^{(l+1)}$ for word $w$. So far, we complete the convolution from $l$-th to $(l+1)$-th layer. For simplicity, we summarize aggregation at Eq. 3–5 by

$$\mathbf{z}_d^{(l+1)} = \text{AGG}(\mathbf{W}_1^{(l+1)}\mathbf{z}_d^{(l)}, \mathbf{W}_2^{(l+1)}\mathbf{z}_w^{(l)}|w \in \mathcal{S}_d), \quad \mathbf{z}_w^{(l+1)} = \text{AGG}(\mathbf{W}_2^{(l+1)}\mathbf{z}_w^{(l)}, \mathbf{W}_1^{(l+1)}\mathbf{z}_d^{(l)}|\forall d : w \in \mathcal{S}_d). \tag{6}$$

We repeat Eq. 6 for maximum $L$ layers and obtain $K$-dimensional topics $\mathbf{z}_d = \mathbf{z}_d^{(L)}$ for document $d$ and $\mathbf{z}_w = \mathbf{z}_w^{(L)}$ for word $w$. The complete encoder is Eq. 7. $\theta$ is the set of all encoding parameters.

$$\mathbf{z}_d, \mathbf{z}_w = f_\theta(\mathbf{z}_d^{(l=0)}, \mathbf{z}_w^{(l=0)}|d \in \mathcal{D}, w \in \mathcal{V}). \tag{7}$$

## 3.2 Missing Semantics Prediction with Contrastive Learning

A short document with few words leaves some content poorly described, resulting in incomplete topic distribution $\mathbf{z}_d$. As a toy example, document B at Fig. 1(c) contains limited words, e.g., gene and clone, and leaves other contents, e.g., protein and cell, uncovered. We aim to complement the topics of short documents. For a document $d$, regardless of long or short, we complement its semantics by

$$\mathbf{z}_d^* = \mathbf{z}_d + \mathbf{m}_d. \tag{8}$$

We name $\mathbf{m}_d \in \mathbb{R}^K$ *missing semantics* of document $d$. If $d$ is a long document with complete semantics, $\mathbf{m}_d$ is a zero vector. A function $g_\mu$ predicts missing semantics at Eq. 9 with topic distributions of $d$ and its support words as inputs. We will elaborate the design of function $g_\mu$ shortly.

$$\mathbf{m}_d = g_\mu(\mathbf{z}_d, \mathbf{z}_w|w \in \mathcal{S}_d). \tag{9}$$

**Contrastive Learning.** Long documents contain relatively more sufficient word co-occurrences than short documents. Thus, we learn missing semantics prediction function $g_\mu$ on long documents, and then transfer the learned semantic knowledge to complement short documents. On one hand, a long document $d$ with enough content does not need semantic complement. Thus we have below constraint

$$\mathbf{m}_d \to \mathbf{0} \quad \text{where} \quad d \in \mathcal{D}_{\text{long}}. \tag{10}$$

On the other hand, although we aim to transfer the semantic knowledge from long to short documents, there does not exist a one-to-one correspondence in corpus $\mathcal{D}$. As a result, we may transfer semantics of a long document (e.g., machine learning concepts) to a short one describing completely distinct content (e.g., biology). To overcome this limitation, we introduce another constraint with contrastive objective. For a long document $d$ with support words $\mathcal{S}_d$, we randomly hide a proportion of words to mimic a short document, denoted as $d'$ with remaining observed words $\mathcal{S}_{d'} \subset \mathcal{S}_d$ and length $l_{d'} < \mathcal{L}$. For both long text $d$ and its short version $d'$, we predict their missing semantics by Eq. 9, and obtain $\mathbf{m}_d$ and $\mathbf{m}_{d'}$, respectively. $\mathbf{m}_{d'}$ should complement the previously hidden semantics, i.e., $\mathcal{S}_d - \mathcal{S}_{d'}$,

$$\mathbf{z}_{d'} + \mathbf{m}_{d'} \to \mathbf{z}_d + \mathbf{m}_d \quad \Rightarrow \quad \mathbf{z}_{d'}^* \to \mathbf{z}_d^* \quad \text{where} \quad d \in \mathcal{D}_{\text{long}}. \tag{11}$$

Together with above Eq. 10, which forces $\mathbf{z}_d^* = \mathbf{z}_d + \mathbf{m}_d$ to approach $\mathbf{z}_d$, Eq. 11 actually has

$$\mathbf{z}_{d'} + \mathbf{m}_{d'} \to \mathbf{z}_d \quad \Rightarrow \quad \mathbf{m}_{d'} \to \mathbf{z}_d - \mathbf{z}_{d'} \quad \text{where} \quad d \in \mathcal{D}_{\text{long}}. \tag{12}$$

To summarize, we arrive at the following constraint loss.

$$\mathcal{L}_{\text{con}}(d) = -\log \sigma(\cos(\mathbf{m}_{d'}, \mathbf{z}_d - \mathbf{z}_{d'})) \quad \text{where} \quad d \in \mathcal{D}_{\text{long}}. \tag{13}$$

$\sigma(x) = \frac{1}{1+\exp(-x)}$ is sigmoid function, and $\cos(\cdot, \cdot)$ is cosine similarity. We here use cosine similarity, mainly due to its superior performance on our datasets. Besides cosine, other similarity metrics, such as inner product and Euclidean distance, are also possible, depending on different datasets.

**Missing Semantics Prediction.** We now define the function of missing semantics prediction $g_\mu$. Given topic distributions of a document and its observed support words as input, a desirable function should aggregate them and output a single missing semantics vector. We propose two alternatives.

1. *GNN function.* The first is to implement $g_\mu$ using a graph neural network, see Fig. 2(b).

$$\mathbf{m}_d = g_\mu(\mathbf{z}_d, \mathbf{z}_w | w \in \mathcal{S}_d) = \text{AGG}(\mathbf{U}_1 \mathbf{z}_d, \mathbf{U}_2 \mathbf{z}_w | w \in \mathcal{S}_d). \tag{14}$$

A corpus $\mathcal{D}$ contains documents with diverse themes. For example, some documents discuss machine learning, while others describe biology. However, the assumption of corpus-level shared parameter $\mathbf{U}_1$ and $\mathbf{U}_2$ is not flexible to model diverse documents for missing semantics prediction, since different documents may have distinct optimal parameters, which are sometimes even in opposing direction. As a result, the predicted missing semantics $\mathbf{m}_d$ centers its mass around the most frequent topics and leaves other distinct topics uncovered. We seek to *personalize* $\mathbf{U}_1$ and $\mathbf{U}_2$ for each document $d$ to recover the distinct missing semantics. Formally, we introduce a function $\phi$, which transforms $\mathbf{U}_1$ and $\mathbf{U}_2$ to document-specific parameters $\mathbf{U}_{d,1}$ and $\mathbf{U}_{d,2}$ by scaling and shifting. Taking $\mathbf{U}_1$ as example,

$$\mathbf{U}_{d,1} = \phi(\mathbf{U}_1, \mathbf{z}_d, \mathbf{z}_w | w \in \mathcal{S}_d) = \mathbf{U}_1 \odot \left[ (\boldsymbol{\gamma}_d + \mathbf{1})_{\times K} \right] + \left[ (\boldsymbol{\beta}_d)_{\times K} \right]. \tag{15}$$

$\boldsymbol{\gamma}_d$ and $\boldsymbol{\beta}_d$ are document-specific vectors for scaling and shifting shared parameter $\mathbf{U}_1$. $\odot$ is element-wise product. $[(\mathbf{x})_{\times K}]$ is a matrix with $K$ identical column vector $\mathbf{x}$. $\mathbf{1}$ is a vector of ones, ensuring the scaling matrix centers around one. $\mathbf{U}_{d,2}$ is similarly defined. Eq. 15 allows each document $d$ to have its own parameters, while all documents still share the common knowledge. Similar documents scale and shift $\mathbf{U}_1$ and $\mathbf{U}_2$ to similar directions. Different documents push them to distinct directions.

We define scaling $\boldsymbol{\gamma}_d$ and shifting $\boldsymbol{\beta}_d$, parameterized by topics of document $d$ and its support words.

$$\boldsymbol{\gamma}_d = \tanh(\mathbf{W}_\gamma \mathbf{z}_d + \mathbf{W}'_\gamma \bar{\mathbf{z}}_{\mathcal{S}_d}), \quad \boldsymbol{\beta}_d = \tanh(\mathbf{W}_\beta \mathbf{z}_d + \mathbf{W}'_\beta \bar{\mathbf{z}}_{\mathcal{S}_d}). \tag{16}$$

$\bar{\mathbf{z}}_{\mathcal{S}_d} = \frac{1}{|\mathcal{S}_d|} \sum_{w \in \mathcal{S}_d} \mathbf{z}_w$ is the average of $d$'s words. In summary, we use Eq. 14 to predict $d$'s missing semantics, except that shared parameters $\mathbf{U}_1$ and $\mathbf{U}_2$ are replaced by $d$-specific ones, $\mathbf{U}_{d,1}$ and $\mathbf{U}_{d,2}$.

2. *Clustering function.* We propose an alternative method by semantic clustering to recover distinct missing semantics, Fig. 2(c). Documents with similar content fall into related clusters, while unique documents belong to different ones. If we assign each cluster a set of parameters for missing semantics prediction, similar documents would recover their own distinct topics. Specifically, we introduce $R$ centroids $\{\mathbf{c}_r\}_{r=1}^R$, each corresponding to one cluster. Given topic distributions of document $d$ and its support words, we first evaluate the assignment probability between document $d$ and each cluster by

$$\Pr(r) = \text{softmax}(-\frac{1}{2}||\mathbf{h}_0 - \mathbf{c}_r||_2^2) = \frac{\exp(-\frac{1}{2}||\mathbf{h}_0 - \mathbf{c}_r||_2^2)}{\sum_{r'=1}^R \exp(-\frac{1}{2}||\mathbf{h}_0 - \mathbf{c}_{r'}||_2^2)}. \tag{17}$$

$\mathbf{h}_0 = [\mathbf{z}_d || \bar{\mathbf{z}}_{\mathcal{S}_d}]$ is the concatenation of $\mathbf{z}_d$ and $\bar{\mathbf{z}}_{\mathcal{S}_d}$. We then assign parameters to each cluster,

$$\mathbf{m}_d = \mathbf{h}_1 = \sum_{r=1}^{R} \Pr(r) \times \text{ReLU}(\mathbf{W}_r \mathbf{h}_0 + \mathbf{b}_r). \tag{18}$$

Therefore, related documents obtain similar clustering probabilities and missing semantics.

Above process is a flat clustering. Our function can be extended to multiple clustering layers. Each layer $s$ consists of $R^{(s)}$ centroids. After obtaining the output from previous layer, $\mathbf{h}_{s-1}$, we repeat Eq. 17–18 by replacing $\mathbf{h}_0$ with $\mathbf{h}_{s-1}$, and obtain the output of the current layer $s$, i.e., $\mathbf{h}_s$. For maximum $S$ layers, we get $\mathbf{m}_d = \mathbf{h}_S$. We leave adaptive learning of number of clusters $R$ as future work.

### 3.3 Probabilistic Decoding with Meta-Learning Optimization

After semantic complement, we obtain $\{\mathbf{z}_d^*\}_{d \in \mathcal{D}}$ at Sec. 3.2. We use $\mathbf{Z}_{\mathcal{V}} = \left[\mathbf{z}_{w_1}; \mathbf{z}_{w_2}; ...; \mathbf{z}_{w_{|\mathcal{V}|}}\right] \in \mathbb{R}^{K \times |\mathcal{V}|}$ to represent topic-word distribution, each column $\mathbf{z}_w$ is topic distribution of a word $w$, and each row is to the distribution of a topic over the vocabulary. As in previous topic models [3, 31], we generate the observed support words $\mathcal{S}_d$ by $\hat{\mathbf{d}}_{\mathcal{S}_d} = \sigma(\mathbf{Z}_{\mathcal{V}} \mathbf{z}_d^*)$. Compared to the ground-truth support words $\mathbf{d}_{\mathcal{S}_d}$, we follow [37] and obtain generative loss $\mathcal{L}_{\text{gen}} = ||\mathbf{d}_{\mathcal{S}_d} - \hat{\mathbf{d}}_{\mathcal{S}_d}||_2^2$. However, this loss requires inefficient computation over the whole vocabulary. We instead use negative sampling [27].

$$\mathcal{L}_{\text{gen}}(d) = \sum_{w \in \mathcal{S}_d} \left[ (d_w - \hat{d}_w)^2 + \sum_{m=1}^{M} \mathbb{E}_{w' \sim \Pr_n(w)} (d_{w'} - \hat{d}_{w'})^2 \right]. \tag{19}$$

$M$ is the number of negative samples, and $\Pr_n(w)$ is a noise distribution over vocabulary. $\hat{d}_w = \sigma(\mathbf{z}_d^{*\top} \mathbf{z}_w)$. $d_w = 1$ if $w \in \mathcal{S}_d$, otherwise $d_w = 0$. In addition, if $d$ is a long document $d \in \mathcal{D}_{\text{long}}$, we also created its corresponding pruned short version $d'$ with a subset of support words $\mathcal{S}_{d'} \subset \mathcal{S}_d$ at Sec. 3.2. Although we do not observe the complete support words for $d'$, its complete topic distribution $\mathbf{z}_{d'}^*$ after semantic complement at Eq. 8 should be able to generate the complete support words $\mathcal{S}_d$. Therefore, together with semantic complement constraint at Eq. 13, we arrive at the complete loss.

$$\mathcal{L}(d) = \mathcal{L}_{\text{gen}}(d) + \mathcal{L}_{\text{con}}(d) + \lambda_{\text{reg}} \mathcal{L}_{\text{reg}}(\theta) \tag{20}$$
$$\text{where} \quad \mathcal{L}_{\text{gen}}(d) = \mathcal{L}_{\text{gen}}(d) + \mathbb{I}(d \in \mathcal{D}_{\text{long}}) \lambda_{\text{gen}} \mathcal{L}_{\text{gen}}(d'), \quad \mathcal{L}_{\text{con}}(d) = \mathbb{I}(d \in \mathcal{D}_{\text{long}}) \mathcal{L}_{\text{con}}(d).$$

$\mathcal{L}_{\text{gen}}(d)$ is generative loss, consisting of document $d$ in the corpus $d \in \mathcal{D} = \mathcal{D}_{\text{long}} \cup \mathcal{D}_{\text{short}}$ and the corresponding pruned short version $d'$ (if $d \in \mathcal{D}_{\text{long}}$). $\mathbb{I}(d \in \mathcal{D}_{\text{long}}) = 1$ if $d \in \mathcal{D}_{\text{long}}$, otherwise 0. $\mathcal{L}_{\text{reg}}(\theta)$ is L2 regularizer for encoding parameters $\theta$. $\lambda_{\text{gen}}$ and $\lambda_{\text{reg}}$ are hyperparameters.

**Optimization.** Finally, with the objective of meta-learning at Sec. 2, we reach the optimization:

1. *Local update.* Given topic encoder $f_\theta$ defined at Sec. 3.1, we optimize encoding parameter $\theta$ w.r.t. the generative loss $\mathcal{L}_{\text{gen}}(d)$ on support words $\mathcal{S}_d$ by Eq. 1 and obtain $\theta_d$ for each document.

2. *Global update.* With encoding parameter $\theta_d$, we compute the overall loss $\mathcal{L}(d)$ on query words $\mathcal{Q}_d$ and optimize all parameters $\Phi = \{\theta, \mu\}$, including encoder parameters $\theta$ and parameters $\mu$ of missing semantics prediction function $g_\mu$. $\alpha_1$ and $\alpha_2$ are local and global learning rate, respectively.

$$\Phi^* = \min_\Phi \sum_{d \in \mathcal{D}} \mathcal{L}(\{\theta_d, \mu\}, \mathcal{Q}_d) \quad \Rightarrow \quad \Phi^* = \Phi - \alpha_2 \nabla_\Phi \sum_{d \in \mathcal{D}} \mathcal{L}(\theta - \alpha_1 \nabla_\theta \mathcal{L}_{\text{gen}}(\{\theta, \mu\}, \mathcal{S}_d), \mathcal{Q}_d). \tag{21}$$

With new parameter $\Phi^* = \{\theta^*, \mu^*\}$, we substitute it for $\Phi$ for the next iteration. In contrast to previous topic models that generate the content of given documents only, we further create the short version of long documents for semantic complement, and jointly optimize them. See supplementary materials for learning algorithm.

### 3.4 Extensions with Auxiliary Data

**MCTM with Pretrained Word Embeddings.** Pretrained word embeddings [27, 29] encode word similarity. As in previous works [7, 45], we incorporate them into topic-word distribution $\mathbf{Z}_{\mathcal{V}} = [z_{k,w}]$ to improve topic modeling. We introduce topic embedding $\{\mathbf{t}_k\}_{k=1}^{K}$ and evaluate cosine similarity between topic $k$ and word $w$ by $\cos(\mathbf{t}_k, \mathbf{h}_w)$. We then combine it with topic-word distribution by $z'_{k,w} = \frac{1}{2}(z_{k,w} + \cos(\mathbf{t}_k, \mathbf{h}_w))$ and obtain a new topic-word distribution $\mathbf{Z}'_{\mathcal{V}} = [z'_{k,w}]$ for decoding.

Table 2: Dataset statistics.

| Name | #Documents | #Links | Vocabulary | #Labels | Avg. #words/doc | Std.Dev. of #words/doc |
|---|---|---|---|---|---|---|
| ML | 2,947 | 8,146 | 5,814 | 7 | 66.7 | 34.0 |
| PL | 2,449 | 7,274 | 5,066 | 9 | 66.0 | 36.9 |
| HEP-TH | 20,151 | 234,193 | 5,001 | N.A. | 48.4 | 22.9 |
| Web | 116,544 | 309,499 | 5,021 | N.A. | 34.1 | 70.0 |

**MCTM with Document Network.** A document network (e.g., citation network) reveals semantic similarities between connected documents (cited papers discuss related research). A document's degree or number of links exhibits a long-tail distribution. Some link to many neighbors, others to a few. Previously focus was on *textual* semantic complement. We extend MCTM to model *structural* semantic complement for link-scarce documents. We consider generating both observed words and neighbors as a task $\mathcal{T}_d$. As for words, we split the neighbors $\mathcal{N}(d)$ of a document $d$ into support and query neighbors $\mathcal{T}_d = \{\mathcal{S}_d, \mathcal{S}_{\mathcal{N}(d)}, \mathcal{Q}_d, \mathcal{Q}_{\mathcal{N}(d)}\}$, $\mathcal{S}_d$ and $\mathcal{S}_{\mathcal{N}(d)}$ denote support words and neighbors, respectively, and ditto for query sets. We correspondingly extend three modeling components.

1. *Encoding.* Previously, we inferred topic distribution $\mathbf{z}_d$ of document $d$ by its textual words at Eq. 6. Here, we extend this process by designing a structural convolutional module.

$$\boldsymbol{\kappa}_d^{(l+1)} = \mathrm{AGG}(\mathbf{W}_3^{(l+1)}\boldsymbol{\kappa}_d^{(l)}, \mathbf{W}_3^{(l+1)}\boldsymbol{\kappa}_{d'}^{(l)}|d' \in \mathcal{S}_{\mathcal{N}(d)}). \tag{22}$$

The topic distribution from encoder $f_\theta$ is $\mathbf{z}_d := \frac{1}{2}(\mathbf{z}_d + \boldsymbol{\kappa}_d)$, with both texts $\mathcal{S}_d$ and structure $\mathcal{S}_{\mathcal{N}(d)}$.

2. *Semantics complement.* For a long document $d$, in addition to randomly hiding some words, we also drop some neighbors and create a pruned version $d'$. Now the missing semantics $\mathbf{m}_d$ should contain both textual and structural information. For GNN function $g_\mu$, we extend Eq. 14 by

$$\mathbf{m}_d = g_\mu(\mathbf{z}_d, \mathbf{z}_w, \mathbf{z}_{d'}|w \in \mathcal{S}_d, d' \in \mathcal{S}_{\mathcal{N}(d)}) = \mathrm{AGG}(\mathbf{U}_{d,1}\mathbf{z}_d, \mathbf{U}_{d,2}\mathbf{z}_w, \mathbf{U}_{d,3}\mathbf{z}_{d'}|w \in \mathcal{S}_d, d' \in \mathcal{S}_{\mathcal{N}(d)}). \tag{23}$$

Scaling has extra input, $\boldsymbol{\gamma}_d = \tanh(\mathbf{W}_\gamma \mathbf{z}_d + \mathbf{W}'_\gamma \bar{\mathbf{z}}_{\mathcal{S}_d} + \mathbf{W}''_\gamma \bar{\mathbf{z}}_{\mathcal{S}_{\mathcal{N}(d)}})$, ditto for shifting. $\bar{\mathbf{z}}_{\mathcal{S}_{\mathcal{N}(d)}} = \frac{1}{|\mathcal{S}_{\mathcal{N}(d)}|} \sum_{d' \in \mathcal{S}_{\mathcal{N}(d)}} \mathbf{z}_{d'}$. For Clustering $g_\mu$, we extend input by $\mathbf{h}_0 = [\mathbf{z}_d || \bar{\mathbf{z}}_{\mathcal{S}_d} || \bar{\mathbf{z}}_{\mathcal{S}_{\mathcal{N}(d)}}]$.

3. *Decoding with meta-learning.* In addition to generating support words using complemented $\mathbf{z}_d^*$, we also generate support neighbors. The generative loss is similar to Eq. 19 except that *i)* we replace $d_w$ with $e_{d,d'}$, the ground-truth link between $d$ and $d'$; *ii)* $\hat{e}_{d,d'} = \sigma(\mathbf{z}_d^{*\top}\mathbf{z}_{d'}^*)$. Finally, meta-learning learns how to accurately predict missing semantics $\mathbf{m}_d$ for both textual and structural complement.

## 4 Experiments

The goal of experiments is to evaluate if our model MCTM can improve short text topic modeling through evaluative tasks, e.g., document classification, link prediction, topic analysis.

**Datasets.** Since our model is flexible to incorporate auxiliary data, we rely on four datasets with textual documents, auxiliary word embeddings, and auxiliary network links for experiments. Cora [24] is corpus of academic papers with citations as links. We created two independent datasets, Machine Learning (**ML**) and Programming Language (**PL**). In addition, **HEP-TH** [21] is a corpus of Physics papers with their citations. **Web** [20] is a Web page hyperlink network where each page is a news article, and the hyperlinks connect related articles. See Table 2 for details.

**Baselines.** Since our model has three variants, i.e., MCTM with plain texts, with auxiliary word embeddings, and with auxiliary document networks, we correspondingly compare to three categories of baselines. *i)* **Topic models with plain texts**, ProdLDA [31], WLDA [28], and GATON [37]. They model all documents uniformly without dealing with short texts. We compare to them and show the advantage of MCTM on improving short texts. *ii)* **Topic models with word embeddings**, ETM [8] and NSTM [45]. Since our model is built on top of GATON, we also compare to GATON with word embeddings, denoted as GATON+WE. By comparing to them, we verify the effectiveness of semantic complement meta-learning to further improve topic quality. *iii)* **Topic models with document networks**, RTM [5], Adjacent-Encoder [40], LANTM [35], and GATON+DN, which is

Table 3: Classification accuracy (in percentage) on four subsets of test set with descending length. Best baselines are underlined. We show improvement of MCTM (G) over GATON and best baseline.

| Category | Model | ML | | | | | PL | | | | |
|---|---|---|---|---|---|---|---|---|---|---|---|
| | | Overall | 0-25% | 25-50% | 50-75% | 75-100% | Overall | 0-25% | 25-50% | 50-75% | 75-100% |
| Models with plain text | ProdLDA | 58.5±3.2 | 69.7±1.0 | 59.1±4.6 | 55.8±5.0 | 48.7±4.6 | 45.0±2.1 | 51.4±6.0 | 48.7±3.6 | 41.6±2.3 | 40.1±3.9 |
| | WLDA | 31.3±0.7 | 30.9±2.9 | 31.6±2.2 | 34.3±1.2 | 28.2±2.2 | 33.2±1.8 | 37.4±1.5 | 40.0±5.3 | 28.1±3.9 | 23.8±1.8 |
| | GATON | 60.3±2.0 | 69.3±2.7 | 62.8±2.5 | 57.4±1.8 | 50.8±6.6 | 47.6±1.5 | 53.8±3.1 | 52.4±3.9 | 44.5±4.9 | 39.0±3.4 |
| | MCTM (G) | 67.1±2.1 | 73.7±3.6 | 68.7±4.3 | 66.4±3.0 | 60.1±1.9 | 53.5±0.8 | 59.8±3.5 | 57.3±2.8 | 52.9±5.7 | 45.5±2.0 |
| | MCTM (C) | 66.9±1.4 | 73.6±1.5 | 68.3±2.0 | 65.2±1.9 | 58.9±1.8 | 53.4±0.9 | 60.6±3.1 | 56.3±1.6 | 52.5±2.0 | 42.9±1.3 |
| | improvement | 11.4%* 11.4%* | 6.3%* 5.7%* | 9.4%* 9.4%* | 15.7%* 15.7%* | 18.3%* 18.3%* | 12.3%* 12.3%* | 11.2%* 11.2%* | 9.3%* 9.3%* | 18.8%* 18.8%* | 16.8%* 13.6%* |
| Models with word embeddings | ETM | 50.6±2.2 | 60.4±3.3 | 52.9±2.5 | 48.8±2.1 | 39.4±3.0 | 43.8±2.0 | 48.5±2.8 | 47.9±2.7 | 42.4±1.9 | 35.9±4.1 |
| | NSTM | 45.2±2.6 | 53.6±1.9 | 42.4±7.5 | 43.0±2.3 | 41.1±4.7 | 41.3±3.2 | 47.2±5.3 | 45.0±5.1 | 39.8±4.6 | 33.1±2.4 |
| | GATON+WE | 63.8±1.5 | 72.4±2.3 | 67.0±3.2 | 60.5±2.2 | 54.8±2.3 | 50.2±1.5 | 57.4±2.3 | 53.7±3.3 | 48.7±2.7 | 40.2±3.1 |
| | MCTM+WE (G) | 66.8±2.0 | 72.9±5.1 | 67.7±2.4 | 65.3±3.8 | 61.6±4.9 | 52.7±1.7 | 60.7±6.1 | 53.9±3.1 | 51.8±1.6 | 43.8±2.6 |
| | MCTM+WE (C) | 66.0±1.4 | 70.9±1.9 | 67.1±3.1 | 67.1±2.8 | 58.3±2.7 | 52.1±1.5 | 60.7±2.7 | 53.1±2.8 | 50.5±4.8 | 43.6±3.6 |
| | improvement | 4.6%* 4.6%* | 0.7%* 0.7% | 1.0% 1.0% | 7.9%* 7.9%* | 12.3%* 12.3%* | 5.0%* 5.0%* | 5.7%* 5.7%* | 0.3% 0.3% | 6.3%* 16.3%* | 9.0%* 9.0%* |
| Models with document networks | RTM | 64.2±2.3 | 72.9±3.6 | 71.0±1.7 | 61.5±4.0 | 50.6±3.4 | 53.3±1.1 | 58.7±3.5 | 58.9±3.0 | 52.4±3.4 | 42.6±2.0 |
| | Adj-Enc | 71.0±0.4 | 78.9±0.7 | 74.6±1.9 | 72.4±2.1 | 57.8±1.4 | 60.4±1.1 | 63.6±1.9 | 63.8±1.5 | 62.6±2.1 | 52.1±3.7 |
| | LANTM | 72.1±1.6 | 74.9±3.1 | 77.2±1.3 | 71.6±2.2 | 64.5±4.5 | 60.8±0.9 | 66.5±3.1 | 61.4±1.5 | 62.4±1.7 | 52.4±3.7 |
| | GATON+DN | 67.7±1.2 | 74.7±3.3 | 71.5±3.1 | 67.8±4.2 | 58.5±2.0 | 58.5±2.0 | 65.4±2.0 | 62.2±3.1 | 61.0±1.7 | 44.3±3.8 |
| | meta-tail2vec | 58.7±1.6 | 65.8±4.5 | 62.0±2.9 | 59.0±3.7 | 47.2±4.0 | 44.9±3.0 | 51.9±2.6 | 48.7±3.1 | 46.6±4.1 | 31.4±6.8 |
| | MCTM+DN (G) | 83.3±1.7 | 86.2±2.9 | 82.7±2.1 | 81.9±2.6 | 82.1±2.4 | 72.9±1.0 | 77.2±4.3 | 73.9±3.8 | 71.9±3.9 | 68.1±2.8 |
| | MCTM+DN (C) | 83.0±1.2 | 85.9±0.7 | 82.0±1.3 | 81.2±1.1 | 82.8±3.9 | 71.9±0.7 | 73.7±3.3 | 72.9±3.0 | 71.0±2.1 | 70.0±2.5 |
| | improvement | 22.9%* 15.4%* | 15.3%* 9.2%* | 15.6%* 7.1%* | 20.9%* 13.1%* | 41.1%* 27.3%* | 24.6%* 19.8%* | 18.2%* 16.1%* | 18.7%* 15.7%* | 18.0%* 14.9%* | 53.7%* 29.9%* |

the extension of GATON with document networks. For document network scenario, we include a graph embedding model, meta-tail2vec [23], which uses meta-learning to improve nodes with low degrees, but is not a topic model and ignores variable lengths of node attributes, i.e., texts.

We set $L = 2$ convolutional layers. $\lambda_{\text{gen}} = 2$ and $\lambda_{\text{reg}} = 0.05$. Number of negative samples $M = 5$ and number of semantic clusters $R = 5$. $\mathscr{L}$ is the median length of the corpus. Local and global learning rates are $\alpha_1 = 0.001$ and $\alpha_2 = 0.0005$. We use 300D Glove embeddings. We experiment with 5 independent runs, report mean and std.dev. All the experiments were done on Linux server with a Tesla K80 GPU with 11441MiB.

## 4.1 Quantitative Evaluation

**Document Classification.** Documents from the same category discuss related topics. As in LDA [3], we conduct classification to evaluate topic quality. We split 80% documents for training (10% are for validation). Labels are not involved during training. After convergence, we train a $k$NN classifier ($k = 5$) [2] with training documents and predict the labels of test documents. We set 64 topics. We compare our models within each category of baselines and report classification accuracy at Table 3. We split the test set into four subsets with descending document length and report the result of both overall test and each subset. 0-25% at Table 3 means the subset with the longest 25% test documents. MCTM (G) and MCTM (C) denote our model with GNN and Clustering function, respectively. We use "*" to represent statistically significant improvement with paired t-test at 0.05 significance level.

Our models significantly outperform baselines within each category. We outperform GATON, the best baseline in the plain text and word embedding category, since textual semantic complement improves short texts, and the overall test set is also improved. MCTM (G) performs slightly better than MCTM (C), potentially because GNN recognizes importance of words with attention, while the Clustering function takes simple average. To show our models indeed improve short text quality, we present the improvement of MCTM (G) over both GATON and the best baseline. Our performance generally improves more as the length decreases, which verifies the advantage of semantic complement.

**Topic Coherence.** Each row of topic-word distribution $\mathbf{Z}_{\mathcal{V}} \in \mathbb{R}^{K \times |\mathcal{V}|}$ is the distribution of one topic over the vocabulary, and the key words of this topic correspond to the highest values on this row. As in ProdLDA [31], we evaluate the coherence of key words by Google Web 1T 5-gram Version 1 [9], with NPMI as metric. Table 4 (left) summarizes the results. Topic-word distribution $\mathbf{Z}_{\mathcal{V}}$ is model parameter and is

Table 5: Topic interpretability.

| Topic | Key words of MCTM (G) |
|---|---|
| 1 | variance, probability, generalize, covariance, approximation |
| 2 | non-genetic, rnn, stimulus-response, epistasis, mismatch |

| Topic | Key words of MCTM (C) |
|---|---|
| 1 | scalability, multiprocessor, obviate, compute, algorithm |
| 2 | sphere, tangent, three-dimensional, vector, geometrical |

separate from document length, thus we can not report results of different lengths. LANTM cannot run on large dataset Web. Meta-tail2vec is not a topic model, thus is excluded. Overall, our models outperform baselines on ML and PL and are competitive with the best baseline on HEP-TH and Web. This indicates that our models at least do not hurt topic coherence, but can significantly improve other tasks, e.g., classification. Compared to GATON, our models significantly improve it, verifying the

Table 4: Topic coherence NPMI (left, in percentage) and perplexity (right) at $K = 64$.

| Category | Model | Topic Coherence NPMI | | | | Perplexity | | | |
|---|---|---|---|---|---|---|---|---|---|
| | | ML | PL | HEP-TH | Web | ML | PL | HEP-TH | Web |
| **Models with plain text** | ProdLDA | 6.3±0.2 | 9.4±0.5 | 10.3±0.6 | 16.2±1.4 | 7.19±0.00 | 7.21±0.00 | 7.72±0.00 | 8.34±0.00 |
| | WLDA | 9.7±0.2 | 11.6±0.1 | **13.7±0.4** | **23.9±0.8** | 18.90±0.73 | 19.57±0.30 | 44.31±0.18 | 45.22±0.00 |
| | GATON | 9.9±0.9 | 8.4±1.5 | 8.9±1.5 | 4.8±1.1 | 9.64±0.27 | 9.17±0.10 | 8.79±0.57 | 8.52±0.12 |
| | MCTM (G) | **10.0±1.4** | **12.1±1.2*** | **13.7±1.7** | 13.5±2.5 | 3.81±0.24 | **3.60±0.51*** | **3.98±0.29*** | 3.27±0.41 |
| | MCTM (C) | 9.9±2.0 | 12.0±1.8 | 13.2±2.1 | 16.1±1.1 | **3.76±0.17*** | 3.63±0.20 | 4.12±0.30 | **3.13±0.13*** |
| **Models with word embeddings** | ETM | 5.5±0.1 | 7.7±0.2 | 7.2±0.4 | 16.4±0.7 | 8.67±0.00 | 8.52±0.00 | 8.51±0.00 | 8.52±0.00 |
| | NSTM | 16.0±1.0 | 18.6±0.6 | 18.2±0.5 | **27.9±0.6** | 8.46±0.00 | 8.34±0.00 | 8.39±0.00 | 8.30±0.00 |
| | GATON+WE | 16.1±1.4 | 12.9±1.5 | 16.9±1.1 | 12.4±1.1 | 5.50±0.21 | 5.56±0.48 | 8.36±0.02 | 7.98±0.02 |
| | MCTM+WE (G) | **17.6±1.2*** | 19.1±2.8 | 18.2±1.3 | 23.6±0.4 | **4.43±0.17*** | **4.23±0.56*** | **3.36±0.10*** | 3.18±0.12 |
| | MCTM+WE (C) | 16.8±1.1 | **20.3±1.5*** | **18.7±0.9** | 23.8±1.1 | 4.62±0.16 | 4.62±0.24 | 3.50±0.17 | **3.04±0.12*** |
| **Models with document networks** | RTM | 7.3±0.2 | 8.9±0.5 | 6.6±0.3 | **18.0±0.4** | 8.07±0.01 | 7.93±0.01 | 8.04±0.00 | 8.96±0.13 |
| | Adj-Enc | 8.4±0.4 | 10.5±0.1 | 6.4±0.4 | 7.2±0.5 | 7.41±0.01 | 7.34±0.13 | 7.45±0.19 | 7.65±0.00 |
| | LANTM | 9.9±1.2 | 9.8±0.7 | 10.4±1.5 | N.A. | 8.63±0.00 | 8.48±0.00 | 8.50±0.00 | N.A. |
| | GATON+DN | 10.3±0.7 | 10.7±1.1 | 9.7±0.8 | 7.7±2.0 | 8.58±0.02 | 8.43±0.00 | 8.33±0.01 | 8.13±0.02 |
| | MCTM+DN (G) | **11.9±2.0*** | **11.1±2.0*** | **10.6±1.7** | 15.5±1.1 | 4.07±0.27 | 4.12±0.31 | **3.99±0.40*** | **3.21±0.25*** |
| | MCTM+DN (C) | 11.6±1.5 | 10.3±1.4 | 10.4±1.5 | 17.0±1.1 | **3.41±0.36*** | **3.63±0.46*** | 4.13±0.23 | 3.75±0.43 |

Table 6: Link prediction (in percentage) on overall test set and the shorter half of the test set. Best baselines are underlined. We show the improvement of MCTM (G) over GATON and best baseline.

| Model | ML | | PL | | HEP-TH | | Web | |
|---|---|---|---|---|---|---|---|---|
| | Overall | Short | Overall | Short | Overall | Short | Overall | Short |
| RTM | 71.4±0.9 | 67.0±0.6 | 68.2±0.4 | 62.2±0.3 | 69.7±0.8 | 65.1±0.6 | 69.9±0.1 | 75.3±0.1 |
| Adj-Enc | 88.1±0.2 | 86.2±0.3 | 79.6±0.3 | 73.8±0.2 | 88.9±0.1 | 88.4±0.1 | 82.7±0.1 | 78.5±0.2 |
| LANTM | 76.5±1.2 | 76.1±1.6 | 73.9±0.9 | 70.5±1.1 | 86.6±0.0 | 85.2±0.0 | N.A. | N.A. |
| GATON+DN | 74.2±0.4 | 71.6±0.9 | 71.6±0.8 | 65.8±0.6 | 90.1±0.2 | 89.1±1.2 | 74.3±0.2 | 71.7±0.3 |
| meta-tail2vec | 69.7±2.3 | 66.5±1.5 | 68.7±1.7 | 64.3±1.5 | N.A. | N.A. | N.A. | N.A. |
| MCTM+DN (G) | **94.0±0.5** | 91.9±1.4 | **91.5±0.1** | **90.6±0.8** | 93.9±0.2 | **93.9±0.1** | **83.4±0.1** | **80.5±0.1** |
| MCTM+DN (C) | 93.4±0.5 | **92.7±0.7** | 91.2±0.8 | 90.2±0.9 | 92.5±0.3 | 92.4±0.4 | 80.6±0.2 | 77.5±0.2 |
| improvement | 26.6%* 6.7%* | 28.4%* 6.6%* | 27.9%* 15.0%* | 37.7%* 22.8%* | 4.2%* 4.2%* | 5.4%* 5.4%* | 12.2%* 0.8%* | 12.3%* 2.5%* |

advantage of semantic complement. To understand what topics our models capture, we randomly present two topics with top-5 key words at Table 5. MCTM (G) captures *statistics* and *computational genetics*, while MCTM (C) reveals *scalability* and *geometric learning*.

**Perplexity.** Topic model should generalize to unseen documents. Following [3], we evaluate perplexity. Since perplexity is exponential and varies much w.r.t. its power, we report its power, $-\frac{\log \Pr(\mathcal{D}_{\text{test}})}{\sum_{d \in \mathcal{D}_{\text{test}}} l_d}$ (lower is better). Table 4 (right) reveals that our models generate high likelihood to unseen documents, which we attribute to semantic complement meta-learning module.

**Link Prediction.** A good model should infer similar topics for potentially linked documents. Since we model document network as auxiliary data, we follow RTM [5] and predict links. As in [40], the probability of a link is $p(e_{d,d'}) \propto \exp(-||\mathbf{z}_d^* - \mathbf{z}_{d'}^*||_2^2)$. We predict the links within test documents and compare with the ground-truth links with AUC as metric. Since only the third category (network models) incorporate links, we mainly compare the MCTM+DN version to these network baselines. Table 6 indicates that our models predict links more accurately than baselines. The comparison to network baselines demonstrates the effectiveness of both textual and structural semantic complement.

## 4.2 Model Analysis

To better understand our model, we conduct model analysis here.

**Effect of Semantic Complement.** To see if semantic complement helps short texts, we remove it from the complete model and present classification accuracy at Table 7 (left). We show the plain text results, and put the auxiliary data version in supplementary. Models do better with semantic complement than without. The accuracy on short subset declines more than the overall test set, which reveals that semantic complement improves short texts, and removing it hurts short documents more.

**Effect of Meta-Learning.** To analyze if meta-learning benefits the optimization with a few observed words, we replace it with the commonly used stochastic gradient descend. Table 7 (middle) shows that result drops more on short subset than on the overall test set, which verifies that meta-learning is good at optimization with only a few observed words and improves short text modeling. We further remove both semantic complement and meta-learning, and report the result at Table 7 (right), which presents the worst accuracy. This observation further verifies that both components are important.

Table 7: Effect of semantic complement and meta-learning on document classification on ML.

| Model | Effect of Semantic Complement | | | | Effect of Meta-Learning | | | | Effect of both | |
|---|---|---|---|---|---|---|---|---|---|---|
| | Overall test set | | Short subset | | Overall test set | | Short subset | | Test | Short |
| | with | without | with | without | with | without | with | without | without | without |
| MCTM (G) (decline) | **67.1±2.1** | 58.2±2.4 (13.3%*) | **60.1±1.9** | 50.8±3.8 (15.5%*) | **67.1±2.1** | 65.9±1.3 (1.8%) | **60.1±1.9** | 57.0±3.5 (5.2%*) | 52.1±4.1 (22.4%*) | 44.5±5.5 (26.0%*) |
| MCTM (C) (decline) | **66.9±1.4** | 56.7±3.1 (15.2%*) | **58.9±1.8** | 49.4±2.9 (16.1%*) | **66.9±1.4** | 65.6±1.0 (1.9%) | **58.9±1.8** | 56.1±2.4 (4.8%*) | 52.5±2.4 (21.5%*) | 46.9±4.8 (20.4%*) |

**Effect of Scaling and Shifting.** The GNN version of our model uses scaling-and-shifting method to recover distinct topics. To test its usefulness, we disregard it and summarize classification accuracy on ML at Table 8 (left). Removing scaling and shifting leads to worse performance, since we can not personalize shared parameters to each document to recover its distinct missing topics for complement.

**Effect of Clustering.** We set the number of clusters $R$ to 1, all documents share the same parameters for missing semantics prediction with no clustering. Table 8(right) concludes that clustering is helpful to share common semantics for related documents and distinguish documents of different clusters.

Table 8: Effect of scaling-and-shifting and clustering.

| Model | Scaling and Shifting | | Effect of Clustering | |
|---|---|---|---|---|
| | with | without | with | without |
| MCTM | **67.1±2.1** | 65.7±2.9 | **66.9±1.4*** | 62.7±1.6 |
| MCTM+WE | **66.8±2.0** | 65.9±1.0 | **66.0±1.4*** | 62.4±1.9 |
| MCTM+DN | **83.3±1.7** | 82.4±1.4 | **83.0±1.2*** | 81.0±1.3 |

# 5 Related Work

Traditional topic models are graphical models [3, 13]. Recent ones are neural models [26], mainly variants of VAE [17]. ProdLDA [31] and DVAE [4] analyze Dirichlet prior. WHAI [43] looks into Gamma prior. The variational divergence metric also attracts attention. WLDA [28] and WAE [32] apply mean maximum discrepancy and generative adversarial network [12]. KATE [6] increases topic sparsity. More recently, topic models are built with graph neural networks [18], e.g., GATON [37], GraphBTM [46], etc. They average losses of all documents regardless of lengths. MCTM designs semantic complement to predict missing semantics for short texts to improve topic quality.

To alleviate scarce word co-occurrences of short texts, existing works leverage auxiliary data, such as word embeddings [27, 29]. Graphical models include GLDA [7], GPUMM [22], LCTM [14], MetaLDA [44], WDL [36], and OTLDA [15]. Neural models include ETM [8] and NSTM [45]. Another category improves short text modeling with document networks, e.g., citation network. RTM [5] and NetPLSA [25] are graphical models. Neural models include NRTM [1], Adjacent-Encoder [40], LANTM [35], NetDTM [42], SemiVN [41]. In contrast, our semantic complement module is agnostic to auxiliary data.

Here, we also briefly review meta-learning works. One category is metric-based, e.g., ProtoNet [30] and MatchingNet [33], which learn a metric function over tasks. Gradient-based meta-learning works optimize model parameters for quick adaptation to new tasks [10, 16, 11, 34, 19, 38, 39].

# 6 Conclusion

We improve short text topic modeling with semantic complement meta-learning. We complement the semantics for short documents by contrastive learning and design two alternatives for missing semantics prediction. Meta-learning helps to optimize and predict the missing semantics. Experiments on document classification, topic coherence, perplexity, and link prediction verify the effectiveness of our model. One limitation is to assume a variable-length corpus with both long and short documents for semantic transfer. We also assume the content is truthful. If the corpus is infiltrated by fake news, those may appear in some topics. A future work is exploring *online learning* to speed up training.

# Acknowledgement

This research/project is supported by the National Research Foundation, Singapore under its AI Singapore Programme (AISG Award No: AISG2-RP-2021-020). Hady W. Lauw gratefully acknowledges the support by the Lee Kong Chian Fellowship awarded by Singapore Management University.

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
