# Meta-Complementing the Semantics of Short Texts in Neural Topic Models – Supplementary Materials

**Delvin Ce Zhang**
School of Computing and Information Systems
Singapore Management University
Singapore 178902
cezhang.2018@smu.edu.sg

**Hady W. Lauw**
School of Computing and Information Systems
Singapore Management University
Singapore 178902
hadywlauw@smu.edu.sg

## 1   Training Algorithm

In this section, we summarize the learning process into the training algorithm at Algo. 1.

## 2   Additional Experiments

### 2.1   Different Number of Topics for Document Classification

In the main paper, we set 64 topics and conduct document classification. Here, we vary the number of topics $K$ from 16 to 256, and summarize classification accuracy on the overall test set at Fig. 1.

Overall, our models perform stably across different number of topics. GATON presents the best results among baseline models in plain text and word embedding category, since it can incorporate higher-order connectivity on the graph by multi-layer convolutions. By comparing to it, our models further improve the performance, due to the advantage of semantic complement meta-learning. When incorporating auxiliary document networks, our models improve GATON more, since both textual and structural semantic complement helps short text topic modeling.

### 2.2   Complete Results of Effect of Semantic Complement and Meta-Learning

Due to space limit, we present the effect of semantic complement and meta-learning for the plain text version of our models in the main paper. Here we show the complete results at Table 1, including the word embedding version and document network version. Similarly, for all three versions, our models without semantic complement or meta-learning deteriorate the results, which verifies the usefulness of semantic complement and meta-learning to improve topic quality. Furthermore, the results decline more on the short subset than on the overall test set, which demonstrates that semantic complement and meta-learning indeed help improve shot text topic modeling, and disregarding them leads to worse performance on short documents.

36th Conference on Neural Information Processing Systems (NeurIPS 2022).

---

**Algorithm 1** Training Process of MCTM

---

**Input**: Corpus $\mathcal{D}$. (Word embeddings $\mathcal{H}$ and document network $\mathcal{E}$, if available.)
**Output**: Topic model $f_\theta$ and semantics complement function $g_\mu$, with parameters $\Phi = \{\theta, \mu\}$.

1: Initialize model parameters $\Phi = \{\theta, \mu\}$.
2: **while** not converged **do**
3:     Sample a minibatch of documents.
    //*Encoding*
4:     **for** each document $d$ in the minibatch **do**
5:         Encoding documents and words by $\mathbf{z}_d, \mathbf{z}_w = f_\theta(\mathbf{z}_d^{(l=0)}, \mathbf{z}_w^{(l=0)})$.
6:         **if** word embeddings are available **then**
7:             $z_{k,w} := \frac{1}{2}(z_{k,w} + \cos(\mathbf{t}_k, \mathbf{h}_w))$.
8:             Obtain new topic-word distribution $\mathbf{Z}_\mathcal{V} = [z_{k,w}]$.
9:         **end if**
10:        **if** document networks are available **then**
11:           $\boldsymbol{\kappa}_d^{(l+1)} = \text{AGG}(\mathbf{W}_3^{(l+1)}\boldsymbol{\kappa}_d^{(l)}, \mathbf{W}_3^{(l+1)}\boldsymbol{\kappa}_{d'}^{(l)} | d' \in \mathcal{S}_{\mathcal{N}(d)})$ for $l = 0, 1, ..., L-1$.
12:           $\mathbf{z}_d := \frac{1}{2}(\mathbf{z}_d + \boldsymbol{\kappa}_d)$.
13:        **end if**
14:     **end for**
    //*Semantic complement*
15:     **for** each document $d$ in the minibatch **do**
16:         **if** $d \in \mathcal{D}_{\text{long}}$ **then**
17:            $d' \leftarrow \text{WordDropout}(d)$, such that $l_{d'} < \mathscr{L}$.
          //*Missing semantics prediction*
18:           Infer missing semantics for pruned short document $d'$ by GNN function or semantics clustering function. $\mathbf{m}_{d'} = g_\mu(\mathbf{z}_{d'}, \mathbf{z}_w | w \in \mathcal{S}_d)$.
19:           Semantic complement $\mathbf{z}_{d'}^* = \mathbf{z}_{d'} + \mathbf{m}_{d'}$.
          //*Constraint loss*
20:           Evaluate constraint loss $\mathcal{L}_{\text{con}}(d)$.
21:         **end if**
22:         **if** $d \in \mathcal{D}_{\text{short}}$ **then**
          //*Missing semantics prediction*
23:           Repeat line 18–19 and obtain complemented $\mathbf{z}_d^*$.
24:         **end if**
25:         Evaluate generative loss $\mathcal{L}_{\text{gen}}(d)$ and complete loss $\mathcal{L}(d)$.
26:     **end for**
    //*Meta-learning optimization*
27:     **for** each document $d$ (including pruned short version $d'$) **do**
28:         Local update for each document $d$ and obtain "personalized" $d$-specific parameter $\theta_d$ .
29:     **end for**
30:     Global update w.r.t. loss for all documents and obtain $\Phi^* = \{\theta^*, \mu^*\}$.
31:     $\Phi \leftarrow \Phi^*$.
32: **end while**

---

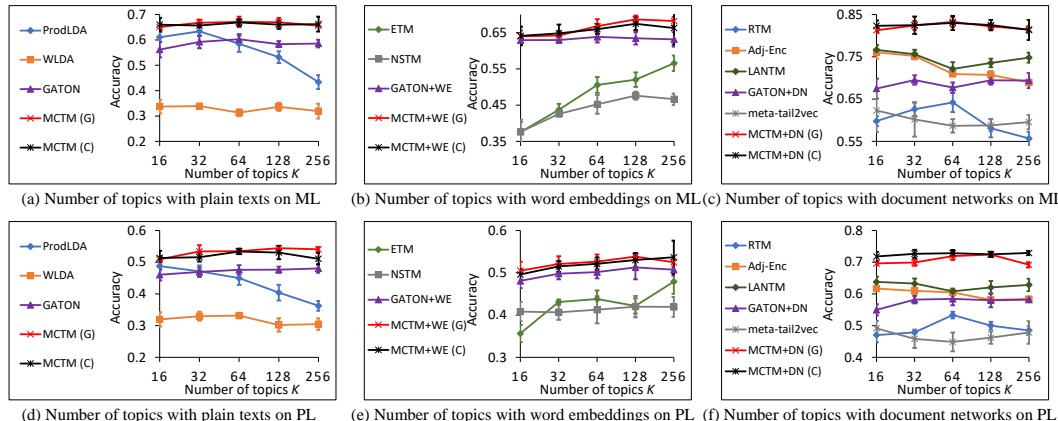

Figure 1: Document classification accuracy with different number of topics on ML and PL.

Table 1: Effect of semantic complement and meta-learning on document classification on ML.

| Model | Effect of Semantic Complement | | | | Effect of Meta-Learning | | | |
|---|---|---|---|---|---|---|---|---|
| | Overall test set | | Short subset | | Overall test set | | Short subset | |
| | with | without | with | without | with | without | with | without |
| MCTM (G) | **67.1±2.1** | 58.2±2.4 | **60.1±1.9** | 50.8±3.8 | **67.1±2.1** | 65.9±1.3 | **60.1±1.9** | 57.0±3.5 |
| (decline) | | (13.3%) | | (15.5%) | | (1.8%) | | (5.2%) |
| MCTM (C) | **66.9±1.4** | 56.7±3.1 | **58.9±1.8** | 49.4±2.9 | **66.9±1.4** | 65.6±1.0 | **58.9±1.8** | 56.1±2.4 |
| (decline) | | (15.2%) | | (16.1%) | | (1.9%) | | (4.8%) |
| MCTM+WE (G) | **66.8±2.0** | 62.6±2.3 | **61.6±1.9** | 53.0±2.4 | **66.8±2.0** | 66.6±0.9 | **61.6±1.9** | 58.9±2.0 |
| (decline) | | (6.3%) | | (14.0%) | | (0.3%) | | (4.4%) |
| MCTM+WE (C) | **66.0±1.4** | 62.1±2.2 | **58.3±2.7** | 52.1±3.0 | **66.0±1.4** | 65.2±1.8 | **58.3±2.7** | 57.7±2.1 |
| (decline) | | (5.9%) | | (10.6%) | | (1.4%) | | (0.9%) |
| MCTM+DN (G) | **83.3±1.7** | 71.6±1.2 | **82.1±2.4** | 68.0±4.2 | **83.3±1.7** | 83.0±1.1 | **82.1±2.4** | 82.0±1.0 |
| (decline) | | (14.0%) | | (17.2%) | | (0.4%) | | (0.1%) |
| MCTM+DN (C) | **83.0±1.2** | 72.5±1.4 | **82.8±3.9** | 69.7±4.3 | **83.0±1.2** | 82.0±1.0 | **82.8±3.9** | 80.0±1.3 |
| (decline) | | (12.7%) | | (15.8%) | | (1.2%) | | (3.4%) |