# OpenReview forum: "Meta-Complementing the Semantics of Short Texts in Neural Topic Models"
_NeurIPS.cc/2022/Conference — NeurIPS 2022 Accept_

### Official Review · Reviewer_yv6N · 2022-06-30

**Rating:** 4
**Confidence:** 4
**Soundness:** 3 good
**Presentation:** 2 fair
**Contribution:** 2 fair

**Summary:**

A number of issues this study are handled are listed as follows.
1. Most of the previous topic models treated the document with various length uniformly.
2. Short documents may only have few words co-occurrences which will make the model hard to identify the corresponding topics.
3. Previous solutions dealt with the short documents require external auxiliary data.

**Questions:**

The proposed method has clear motivation that it wants to treat short documents and the long documents uniquely. Therefore, more semantics can be extracted from the short documents to increase model performance in classification and topic coherence. Meta-learning is also introduced to deal with the limited data. However, the learning criterions used in this work is unclear and the organization is not well adjusted. The figure only delivers limited information. Since the learning process is quite complicated, it is suggested to provide Algorithm.

**Ethics Review Area:**

["I don’t know"]

**Limitations:**

The novelty is not sufficiently justified.

**Strengths And Weaknesses:**

Pros
1. The proposed method showed good performances in their evaluation.
2. Ablation study was conducted.

Cons
1. Content organization was not well adjusted. Figure only showed the limited information and there is no Algorithm. Therefore, it was quite hard for the readers to understand the learning process.
2. The methodology was not well-explained:
  (1) In the contrastive loss, “What is the meaning of contrastive in the loss function?, because there are no positive negative pairs” and “What is the intuition of using cosine similiraity as an input for softmax function?”
  (2) In the generative loss, " what is the function of m?", It just described as number of negative samples, but what is the role of m in the equation?
3. The proposed method was not evaluated on the dataset commonly used in the mentioned previous works, which is 20newsgroup. Therefore, it’s hard to say that the proposed method could outperform the previous solutions.

---

> ### Author Response · Authors · 2022-08-02
> **Response to Reviewer yv6N**
>
> We thank the reviewer for the feedback. Below we address the concerns and questions.
>
> ---
>
> R1: Content organization was not well adjusted. Figure only showed the limited information and there is no Algorithm. Therefore, it was quite hard for the readers to understand the learning process. It is suggested to provide Algorithm.
>
> A1: We thank the reviewer for the suggestion. We summarize the learning process and provide a learning algorithm at Section 2 of the newly uploaded supplementary material. We will move it to the main paper upon acceptance.
>
> ---
>
> R2: In the contrastive loss, what is the meaning of contrastive in the loss function? because there are no positive negative pairs
>
> A2: Contrastive learning appears at Section 3.2 where we design a constraint term for missing semantics prediction. Specifically, as explained at line 121-122, although we aim to transfer the semantic knowledge from long to short documents, there does not exist a one-to-one (long-to-short) correspondence in the corpus. To solve this problem, at line 125, we explain the method of hiding a proportion of words from a long document $d$ to generate a pruned short document $d’$. We use this pair of documents, $d$ and its __corresponding__ short version $d’$, to represent a __contrast__ of a __long-vs-short__ document pair to learn the missing semantics prediction function.
>
> ---
>
> R3: What is the intuition of using cosine similarity as an input for softmax function?
>
> A3: At Eq. 13, we aim to maximize the similarity between $m_{d’}$ and $z_d-z_{d’}$. There are several possible similarity metrics, such as cosine similarity, Euclidean distance, inner product, etc. We discover that cosine similarity generally performs well on our datasets. We thus use cosine similarity. Other metrics are also possible, depending on the datasets. We thank the reviewer for reminder. We add this explanation to the newly uploaded revision paper at line 131-133.
>
> ---
>
> R4: In the generative loss, " what is the function of m?", It just described as number of negative samples, but what is the role of m in the equation?
>
> A4: As explained at line 171 in the newly uploaded revision paper, we follow previous works (GATON [37]) and optimize generative loss by minimizing sum-of-squares error between ground-truth Bag-of-Words vector and the generated content vector. The ground-truth Bag-of-Words vector contains __all__ __the__ __words__ in the vocabulary. Some words are positive integers (word occurrences), while most words do not appear in the document and have 0 value in Bag-of-Words vector. However, the generative loss at line 171 requires the sum of squares for all the words where each word is obtained by graph convolutional encoder. Line 171 is computationally expensive, especially when the vocabulary size is large. To speed up the training, as explained at line 172, we follow [27] and use negative sampling at Eq. 19. Specifically, for each word with positive occurrence, we randomly sample $M$ words that do not appear in the document to represent all the words with 0 value. Different training epochs sample different set of negative words. __The__ __role__ __of__ $M$ is to use a set of randomly sampled non-occurrence words to represent all the non-occurrence words in the ground-truth Bag-of-Words vector. __The__ __value__ __of__ $M$ is negative sampling rate, i.e., how many non-occurrence words we use for each positive occurrence word for optimization.
>
> ---
>
> R5: The proposed method was not evaluated on the dataset commonly used in the mentioned previous works, which is 20newsgroup. Therefore, it’s hard to say that the proposed method could outperform the previous solutions.
>
> A5: As explained at line 224-226 in the newly uploaded revision paper, since our model has extensions for word embeddings and __document__ __networks__, we seek datasets that have __both__ word embeddings and document networks for experiments, so that we can __consistently__ use the same datasets for all evaluation tasks, including network link prediction. However, 20NG does not have document network structure, thus we cannot run any document network models on 20NG or evaluate link prediction task. On the contrary, ML, PL, and HEP-TH are commonly used datasets with required document networks. Many previous works, including RTM, Adj-Enc, and LANTM, also use these datasets to evaluate topic models __with__ __plain__ __text__, including ProdLDA, WLDA, ETM, etc. Thus, for consistency purpose, we use datasets with required document networks for experiments.
>
> ---
>
> Please let us know if you have any further questions or comments.

---

### Official Review · Reviewer_zq8q · 2022-07-12

**Rating:** 7
**Confidence:** 4
**Soundness:** 3 good
**Presentation:** 3 good
**Contribution:** 3 good

**Summary:**

In this paper, the authors address the limitation of previous topic models in modeling short texts. Their analysis demonstrates a decreasing classification accuracy with decreasing text lengths, due to the lack of word co-occurrences in short texts. They propose to complement the semantics of short text, by training a "missing semantic prediction" model on long text. Specifically, a proportion of a long text is masked and the missing semantic prediction model is expected to output a complementary representation ("missing semantics"), which adds to the representation of masked long text to reconstruct the representation of the full long text. The authors propose two configurations of the missing semantic prediction model, which are based on GNN and clustering respectively, and GNN seems to be a slightly better choice. It can also be extended to leverage external data, such as pre-trained word embeddings or document networks for further improvement. Experimental results show significant improvement in document classification, topic coherence, perplexity, and link prediction, for both long and short texts.

**Questions:**

1. The authors choose to complement document representations. However, it might be worth exploring a textual level complement, such as using a language model to generate extra sentences on short texts for richer semantics.
2. Is the constraint in formula (10) met? In other words, will the model still complement relatively long texts, or keep them unchanged?
3. It is quite surprising to observe a large improvement in long texts (0-25%) classification as well. What could be the reason for improvement? Will there be such an improvement if we take a look at more extremely long texts (0-5%)?

**Limitations:**

The authors did not include a discussion on limitations or ethical impact.

**Strengths And Weaknesses:**

Strengths:
1. The paper is well motivated with quantitative analysis to demonstrate the disadvantage of previous topic models in handling short texts.
2. The method is novel in complementing the semantics of short text to overcome the lack of word co-occurrences in short texts.
3. Overall, the paper is well written with a comprehensive explanation of the method and formulations.
4. The experimental results show significant improvement, and the ablation studies are sufficient to justify the design choices.

Weaknesses:
1. Some important hyperparameter details are missing, such as the threshold for defining short text and long text, the percentage of masking long text, negative sampling rate, etc.
2. The explanation of the meta-learning part is unclear. How do you optimize formulas 1 and 2? Also, it might be better to merge the paragraphs on meta-learning at the end of Section 2 and Section 3.3 for easier understanding.

---

> ### Author Response · Authors · 2022-08-02
> **Response to Reviewer zq8q**
>
> We thank the reviewer for the feedback. Below we address the concerns and questions.
>
> ---
>
> R1: Some important hyperparameter details are missing.
>
> A1: At line 238-239, we actually report the threshold of short and long texts is the median length of the corpus. At line 237, the number of negative samples M is 5. For a long document, we randomly hide some words until the length of the remaining masked document is lower than the threshold $l_{d'}<\mathscr{L}$. We also submitted the code with more details for your reference.
>
> ---
>
> R2: The explanation of the meta-learning part is unclear. How do you optimize formulas 1 and 2?
>
> A2: We base the meta-learning on the widely used MAML. Here we explain more about Eq. 1-2. We will add below explanation to the camera-ready version.
>
> During training, MAML has two steps, local and global update.
>
> 1. Local update. Given a topic model $f_\theta$ with parameter $\theta$, MAML updates $\theta$ on the support words $S_d$ of __each__ individual document $d$ by Eq. 1 and obtain $\theta_d$. Note that MAML optimizes $\theta$ on each individual document, so that $\theta_d$ is document $d$-specific, because different documents have different generative loss. As a result, different documents have different $\theta_d$, but they obtain $\theta_d$ from the __same__ global parameter $\theta$. This is why this step is called local update ($\theta$ is locally updated w.r.t. each individual document). The purpose of local update in MAML is to train the model $f_\theta$ to quickly adapt to each document by Eq. 1.
>
> 2. Global update. Since different documents now have their own personalized (or local) parameter $\theta_d$, MAML aims to see if above local update indeed optimizes the parameter $\theta$ to the correct direction for each document, so that local parameter $\theta_d$ is effective to infer document $d$’s topic distribution. This purpose is achieved by global update Eq. 2, where MAML evaluates the loss of local parameter $\theta_d$ on query words $Q_d$ to see if local parameter $\theta_d$ indeed generates an effective topic distribution and low generative loss. Since $\theta_d$ is obtained by global parameter $\theta$ at Eq. 1, the optimization of local parameter $\theta_d$ is actually the optimization of global parameter $\theta$. Each training epoch in our model alternates local and global updates to optimize parameters until convergence.
>
> ---
>
> R3: It might be worth exploring a textual level complement, such as using a language model to generate extra sentences on short texts for richer semantics.
>
> A3: Using language model to generate complementary text may suffer from below problem. Language model is usually pre-trained on external large corpus with __various__ topics and vocabulary, thus the extra sentences it generates to complement short documents are likely to express some topics outside our given corpus, which influences the topic discovery process of a topic model. For example, suppose we aim to discover topics from a machine learning corpus, but the pre-trained language model may generate a lot of biology sentences for a short ML document with content “the evolution of genetic algorithm”. Even though we condition the content generation of language models on the given short texts, the scarce word co-occurrences of short texts may not accurately allow language models to complement the correct content. Consequently, we cannot control the generation process. In contrast, our model is self-contained within the given training corpus. We complement the topics of short texts using long documents in the same corpus.
>
> ---
>
> R4: Is the constraint in formula (10) met? In other words, will the model still complement relatively long texts, or keep them unchanged?
>
> A4: In our implementation, we keep topic distributions of long documents unchanged.
>
> ---
>
> R5: It is quite surprising to observe a large improvement in long texts (0-25%) classification. What could be the reason for improvement? Will there be such an improvement if we take a look at more extremely long texts (0-5%)?
>
> A5: i) The improvement of long documents is mainly due to the effectiveness of meta-learning optimization, which is good at improving performance when observed words are finite. Even though long documents contain relatively more words than short ones, their content is still finite, and meta-learning still has some effect on long documents. ii) We used the submitted code to produce classification accuracy of top-5% long documents, and observed less improvement over GATON than top-25% documents. The __improvement__ is due to the effect of meta-learning. The __less__ improvement than top-25% documents is because of more observed words.
>
> ---
>
> R6: The authors did not include a discussion on limitations or ethical impact.
>
> A6: At line 337, we actually have a discussion on limitations. At line 338-339, we also have a discussion on ethical impact.
>
> ---
>
> Please let us know if you have any further questions or comments.

---

### Official Review · Reviewer_kUTk · 2022-07-14

**Rating:** 6
**Confidence:** 3
**Soundness:** 3 good
**Presentation:** 3 good
**Contribution:** 3 good

**Summary:**

This paper focuses on improving topic modeling performance on short documents. Short documents have fewer word co-occurrences that results in worse topic quality. The paper assumes a corpus of documents of various lengths and leverage the longer document to complement the semantics of the short text. They intend to improve the performance on short text without hurting the performance on the longer text. Their model can also work with auxiliary data such as pre-trained word embeddings and the document connectivity structure. Experiments demonstrate improvement of classification accuracy on Cora-ML and Cora-PL, with shorter documents benefit more. Topic coherence and perplexity is evaluated and reported on Cora-ML, Cora-PL, HEP-TH, and Web. The proposed models outperform other methods on Cora-ML and Cora-PL and are similar to baseline methods on HEP-TH and Web.

**Questions:**

1. If I read correctly, in Table 2, it seems MCTM with plain texts is performing better than MCTM with auxiliary word embeddings. I'm wondering if there is any explanation of why auxiliary information (word embeddings) is not helping. (The presentation of Table 2 could be improved by adding the description of each group (Topic models with plain text, etc.)

2. I'm wondering if Table 6 can be re-organized and adding the number of without both semantic complement and meta-learning. It seems the effect of semantic complement is bigger.

3. I wonder if semantic complement could be viewed as a prediction/generation problem (given z_d', predict m_d') and utilize the pre-trained large language model for such prediction.

**Limitations:**

I'm assuming that if the length of the document is not uniform across the topics, for example, certain topics only contains short documents, it might be a problem for the proposed method. I'm wondering if the author has done analysis regarding this.

**Strengths And Weaknesses:**

The paper addresses a focused problem: how to improve the performance on short document when doing topic modeling, which relies on the co-occurrence of words. The semantics transfer and complement seems to have good improvement on the performance.

---

> ### Author Response · Authors · 2022-08-02
> **Response to Reviewer kUTk**
>
> We thank the reviewer for the feedback. Below we address the concerns and questions.
>
> ---
>
> R1: If I read correctly, in Table 2, it seems MCTM with plain texts is performing better than MCTM with auxiliary word embeddings. I'm wondering if there is any explanation of why auxiliary information (word embeddings) is not helping.
>
> A1: Table 2 shows document classification accuracy. One possible reason is that since topic distributions of short documents are already complemented by our model, their topic distributions contain enough semantics to be classified well. Even though auxiliary word embeddings are further available, they cannot improve document classification too much, due to the already abundant semantics. However, pre-trained word embeddings capture word similarities, and Table 3 (left) shows that word embeddings significantly help improve topic coherence. This means auxiliary word embeddings are __still__ __helpful__ to improve topic modeling quality, and their effectiveness is mainly presented by other evaluation metrics, such as topic coherence.
>
> ---
>
> R2: The presentation of Table 2 could be improved by adding the description of each group (Topic models with plain text, etc.
>
> A2: We thank the reviewer for the suggestion. We improved this in the newly uploaded revision paper.
>
> ---
>
> R3: I'm wondering if Table 6 can be re-organized and adding the number of without both semantic complement and meta-learning. It seems the effect of semantic complement is bigger.
>
> A3: Table 6 aims to verify the effectiveness of semantic complement (left) and meta-learning (right), respectively. One correct method for verification is to __respectively__ remove each component from the complete model, and compare the result before and after removal. If the performance drops, it means the component is indeed useful. Table 6 is already enough to verify the effectiveness of these two components. But for completeness, we still add the result of the model without both components to Table 6 of the newly uploaded revision paper. The result shows that the model without both components presents the lowest performance, which further enhances the effectiveness of both designs.
>
> Based on the results at Table 6, we observe that the effect of semantic complement is bigger than meta-learning. The possible reason is that meta-learning is just an alternative __optimization__ __method__ of SGD and does not directly touch topic distributions. On the contrary, semantic complement aims to solve word sparsity problem by __directly__ adding missing semantics to topic distribution, which can improve the quality of topic distributions better than meta-learning.
>
> ---
>
> R4: I wonder if semantic complement could be viewed as a prediction/generation problem (given z_d', predict m_d') and utilize the pre-trained large language model for such prediction.
>
> A4: Using language model to generate complementary text may suffer from below problem. Language model is usually pre-trained on external large corpus with __various__ topics and vocabulary, thus the extra semantics it generates to complement short documents are likely to express some topics outside our given corpus, which influences the topic discovery process of a topic model. For example, suppose we aim to discover topics from a machine learning corpus, but the pre-trained language model may generate a lot of biology sentences for a short ML document with content “the evolution of genetic algorithm”. Even though we condition the content generation of language models on the given short texts, the scarce word co-occurrences of short texts may not accurately allow language models to complement the correct content. Consequently, we cannot control the generation process. In contrast, our model is self-contained within the given training corpus. We complement the topics of short texts using long documents in the same corpus.
>
> ---
>
> R5: I'm assuming that if the length of the document is not uniform across the topics, for example, certain topics only contains short documents, it might be a problem for the proposed method. I'm wondering if the author has done analysis regarding this.
>
> A5: Documents are usually represented by a mixture of __related__ __topics__, i.e., documents’ topic distributions are not one-hot encoding, but instead, a probability distribution over several related topics. Thus, even though some topics do not have long documents, the long documents from related topics can still help improve short texts. Furthermore, if we really cannot obtain enough long documents from related topics and cannot complement short texts, we at least __do__ __not__ __hurt__ the quality of short text topic modeling. If related long documents are available, we can further improve short texts. This is the purpose of our model.
>
> ---
>
> Please let us know if you have any further questions or comments.

---

### Official Review · Reviewer_hYpp · 2022-07-16

**Rating:** 8
**Confidence:** 4
**Soundness:** 4 excellent
**Presentation:** 3 good
**Contribution:** 4 excellent

**Summary:**

This paper proposes a meta-complement topic model (MCTM) to improve the quality of discovered topics in short texts. The key idea is to transfer the semantic knowledge learned on long documents to short texts via meta-learning. Some previous studies rely on external data, such as pre-trained word embeddings and document citation networks, to mitigate the scarcity of word co-occurrences in short texts. In contrast, this work proposes to solve this problem within the corpus itself. The authors also show how to generalize their framework when word embeddings or citation networks are available. Experiments on 4 datasets demonstrate the effectiveness of MCTM in text classification, link prediction, topic coherence, and perplexity.

**Questions:**

- Could you conduct t-tests in your result tables and report whether your improvement over baselines/ablation versions is statistically significant or not?

**Limitations:**

The authors mention the technical limitations in Section 6.

They also mention the potential negative societal impact of their work in Section 6.

**Strengths And Weaknesses:**

Strengths:
+ The idea of using meta-learning to deal with short texts in topic modeling is novel and intuitive from my perspective. According to Section 3.4, it is also straightforward to extend the model when external signals are available, indicating the flexibility of MCTM.

+ The empirical study is comprehensive. Besides the comparison of common topic modeling metrics (i.e., NPMI and perplexity), the authors also explore downstream tasks such as document classification and link prediction (given that the model can be extended to utilize document links).

+ Ablation studies demonstrate the contribution of different components in model design.

Overall, this is a solid submission in my view. However, I still need to explain the following minor concerns.

Weaknesses:
- Significance tests are missing in all result tables. It is unclear whether your improvement over baselines/ablation versions is statistically significant or not. In fact, in Tables 3 and 7, some gaps are quite subtle. Please conduct t-tests and report p-values. (Also, please order all tables according to their number. For example, Table 1 should be presented before Table 2.)

- For the evaluation of topic modeling, the adopted metrics are NPMI and perplexity. However, a recent study [1] shows that such metrics may not align well with human judgment. I wonder if more human annotations can be included (e.g., the intrusion test) when evaluating topic qualities.

[1] Is Automated Topic Model Evaluation Broken? The Incoherence of Coherence. NeurIPS'21.

---

> ### Author Response · Authors · 2022-08-02
> **Response to Reviewer hYpp**
>
> We thank the reviewer for the feedback. Below we address the concerns and questions.
>
> ---
>
> R1: Please conduct t-tests and report p-values.
>
> A1: We reported both mean and standard deviation for all the results in our submission, which can reveal the significance of the improvement. But for completeness, we conduct t-test and report p-values in the newly uploaded revision paper. We use “*” to represent statistically significant improvement based on paired t-test at the significance level of 0.05. Overall, most results present statistically significant improvement. For Table 3, our models significantly outperform baselines on ML and PL in most cases, and are competitive with the best baseline on HEP-TH and Web. As explained at line 272 in the newly uploaded revision paper, we at least do not hurt topic coherence, but can significantly improve other tasks, e.g., classification and link prediction. For Table 7, although the difference with and without scaling/shifting is marginal, we still discovered a consistent slight improvement after taking 5 independent and random experiments.
>
> ---
>
> R2: Please order all tables according to their number. For example, Table 1 should be presented before Table 2.
>
> A2: We thank the reviewer for the suggestion. We re-order these two tables in the newly uploaded revision paper.
>
> ---
>
> R3: For the evaluation of topic modeling, the adopted metrics are NPMI and perplexity. However, a recent study [1] shows that such metrics may not align well with human judgment. I wonder if more human annotations can be included (e.g., the intrusion test) when evaluating topic qualities.
>
> A3: After reviewing previous works in topic modeling, we discovered that NPMI and perplexity are still standard and widely used evaluation metrics. All our topic modeling baselines have either or both of them for evaluation. These two metrics are also adopted by many reputable works, including LDA (perplexity), RTM (perplexity), ETM (NPMI), GATON (NPMI), NSTM (NPMI), and ProdLDA (both). Other evaluation metrics are possible, but they are still not yet widely used, and human annotation may also be relatively subjective. Thus, we follow previous works and mainly adopt NPMI and perplexity for topic modeling evaluation.
>
> ---
>
> Please let us know if you have any further questions or comments.

---

### Meta-Review · Area_Chair_tspr · 2022-08-29

**Recommendation:** Accept
**Confidence:** Certain

**Metareview:**

Most of the reviewers were content with the paper and the interactions with the authors.   I was personally convinced by the response given to the fourth reviewer, primarily about the 20NG dataset, and hence suggesting acceptance.

**Award:**

No

---

### Decision · Program_Chairs · 2022-09-14

Accept